# Use of Seawater/Brine and Caliche's Salts as Clean and Environmentally Friendly Sources of Chloride and Nitrate Ions for Chalcopyrite Concentrate Leaching

**Pía Hernández** [1,*] **, Alexis Dorador** [1] **, Monserrat Martínez** [1] **, Norman Toro** [2,3] **, Jonathan Castillo** [4] **and Yousef Ghorbani** [5]

1. Departamento de Ingeniería Química y Procesos de Minerales, Universidad de Antofagasta, Av. Angamos 601, Antofagasta 1270300, Chile; alexisdoradorlopez@gmail.com (A.D.); monserrat.martinez.vergara@ua.cl (M.M.)
2. Faculty of Engineering and Architecture, Universidad Arturo Prat, Almirante Juan José Latorre 2901, Antofagasta 1244260, Chile; ntoro@ucn.cl
3. Departamento de Ingeniería en Metalurgia y Minas, Universidad Católica del Norte, Av. Angamos 0610, Antofagasta 1270709, Chile
4. Departamento de Ingeniería en Metalurgia, Universidad de Atacama, Av. Copayapu 485, Copiapó 1531772, Chile; jonathan.castillo@uda.cl
5. MiMeR—Minerals and Metallurgical Engineering, Luleå University of Technology, SE-971 87 Luleå, Sweden; Yousef.ghorbani@ltu.se
* Correspondence: pia.hernandez@uantof.cl; Tel.: +56-55-2-637-525

**Abstract:** A less harmful approach for the environment regarding chalcopyrite concentrate leaching, using seawater/brine and caliche's salts as a source of chloride and nitrate ions, was investigated. Different variables were evaluated: sulfuric acid concentration, sodium nitrate concentration, chloride concentration, source of water (distilled water, seawater, and brine), temperature, concentrate sample type, nitrate source (analytical grade and industrial salt), and pre-treatment methods in order to obtain maximum copper extraction. All tests were performed at moderate temperatures (≤45 °C) and atmospheric pressure. The leaching system using distilled water, seawater, and brine base media resulted in copper extraction of 70.9%, 90.6%, and 86.6% respectively. The leaching media, with a concentration of 20 g/L Cl⁻, obtained a maximum Cu extraction of 93.5%. An increase in the concentration of $H_2SO_4$ and $NaNO_3$ from 0.5 to 0.7 M, led to an increase in the copper extraction. The use of an industrial salt compared to the analytical salt did not show great variations in the percentage of extraction achieved, which would be a good and cost effective alternative. The increase in temperature from 25 to 45 °C showed a great effect on the copper leaching (of 60% until 90.6%, respectively). The pre-treatment is suggested to increase copper extraction from 60.0% to 71.4%.

**Keywords:** chalcopyrite leaching; seawater; industrial brine; caliche's salts; chloride; nitrate

## 1. Introduction

The grades of copper minerals in mining deposits in Chile are decreasing [1] due to the exploitation of them over time. Moreover, currently in Chile, copper oxide mineral deposits are being depleted due to excessive mining. In the near future, solvent extraction (SX) and electrowinning (EW) plants will become obsolete. In this context, there is a great opportunity to continue investigating new processes in order to continue the operation of SX and EW plants. Furthermore, due to the depth of the deposits, copper sulfides species, especially chalcopyrite, is abundant in almost all copper mining deposits in

Chile. This mineral is processed by the concentration path by flotation and subsequent pyrometallurgy. Flotation consumes a large amount of water (8.15 $m^3$/s, equivalent to 61% of the total consumption of the copper process) and today, the use of seawater is increasing in copper mining companies (23% of water consumption in the copper mining process, corresponds to seawater) [1,2]. Pyrometallurgy must meet increasingly demanding environmental requirements due to the generation of $SO_2$ and arsenic; therefore, the development of sustainable foundry technologies is an issue that involves new innovations and high implementation costs [3].

On the other hand, chalcopyrite leaching presents challenges because this mineralogical species is refractory in sulfate medium, producing a passivation layer around the particle that causes the slow dissolution rate and low copper extraction, especially at room temperature. It needs high process temperatures and oxidizing media such as the use of oxygen, ferric ion, chlorine gas, and nitrate ion, among others [4,5]. Many researchers have shown that a chloride medium favors the dissolution of chalcopyrite in an acidic medium [6–9]. The use of chloride ions helps the formation of the chloro complex of copper (I) and (II) [10,11]. These chloro complexes have the ability to stabilize cuprous ions and in turn, increase the solubility of copper sulfide [9,12]. Comparing a sulfate system with a chloride system, the copper dissolution kinetics are much faster in the chlorinated medium. Increasing the chloride concentration has also been shown to increase copper extraction [6,13–15]. Carneiro and Leão [16] demonstrated that due to the presence of NaCl in the chalcopyrite leaching system, the morphology of the product layer formed around the leached particle was altered, increasing the porosity of the mineral and maximizing the copper dissolution. A chloride media can be provided by the seawater or discard brine from reverse osmosis plants [13–15,17]. Due to the lower availability of water in the country, what is exacerbated in the north of Chile, which concerns the majority of the copper mining company, is that the use of seawater is an alternative to the availability of water resources. In addition, in recent years, several desalination plants have been installed in the north, providing an alternative to use the waste brine from this process [2].

Moreover, oxidant ions are necessary to extract the copper from the chalcopyrite. The use of $NaNO_3$ is an efficient alternative, as exposed by various researchers, by providing the ion ($NO_3^-$) for leaching chalcopyrite due to its high oxidation potential. This salt is very abundant in northern Chile, produced by the caliche industry [18,19], where the exploitation of copper minerals occurs mostly in this area. Sodium nitrate in an acidic medium provides the opportunity for a possible leaching application of many sulfide minerals, including chalcopyrite [20–22]. Caliche contains nitrate, and chloride ions [19,23] which can help chalcopyrite leaching. Several researchers have investigated the positive effect of using nitrate ions on chalcopyrite leaching [24–27]. Habashi [28] determined that in a nitrate medium, the oxidation of metal sulfide with $HNO_3$ can be carried out in two ways. In the first, $NO_3^-$ is the oxidizing agent and during the reaction, it is reduced to $NO_2$ or NO, and in the second case, the oxidant is oxygen, which results from the decomposition of $HNO_3$. Sokić et al. [25] studied the leaching behavior of chalcopyrite concentrate using sodium nitrate and sulfuric acid media. They determined that the copper extraction increased with increasing nitrate and sulfuric acid concentration. Shiers et al. [27] evaluated three types of oxidants for leaching of chalcopyrite: hypochlorous acid, chlorate and nitrate. The results regarding the three systems tested determined that the use of nitrate is the most cost-effective option. Narangarav et al. [26] obtained over 85% of copper extraction from chalcopyrite leaching in an acid-nitrate media at 90 °C. They determined an activation energy of 15.96 kJ/mol, indicating that the kinetics are chemically controlled. Hernandez et al. [20] studied the chalcopyrite leaching from a low-grade copper sulfide ore in a chloride-nitrate-acid media at 45 °C. Copper extraction of 80% was obtained at 45 °C, in stirred leaching after seven days. The chloride ions were beneficial to improve the copper dissolution in the media. Moreover, Hernandez et al. [29] studied the effect of the pre-treatment stage in chalcopyrite leaching in an acid-nitrate-chloride media. Copper extraction of 58.6 % was obtained after 30 days of pre-treatment at 45 °C. Only 22.8% of Cu was obtained at 25 °C using the same experimental conditions of pre-treatment. Castellón et al. [21] proposed an alternative process to chalcopyrite leaching in a nitrate-chloride-acid media and subsequent recovery

of nitrate ions. A maximum of 90.8% Cu was obtained when 0.5 M of $NaNO_3$ and 0.5 M $H_2SO_4$ at 45 °C and $P_{80}$ of 60.7 μm were evaluated in the leaching system.

This study is part of a project aiming to introduce a greener and environmentally friendlier approach for chalcopyrite concentrate leaching using seawater/brine (as a source of chloride) and caliche's salts (as a source of nitrate ions) in an acid media at moderate temperatures (≤45 °C) [20,29,30]. The overall aim was to provide an alternative method to tackle the problems that affects the Chilean copper mining such as the exhaustion of oxidized copper minerals and the scarcity of water. In the spirit of the circular economy and industrial waste valorization, the approach utilizes effective ions to form waste from two other industrial processes, namely seawater desalination (brine) and caliche mineral heap leaching. The behavior of chalcopyrite concentrate leaching was studied in a nitrate-chloride-acid media in order to determine the effect of several variables: sulfuric acid concentration (0.5 and 0.7 M), sodium nitrate concentration (0.5 and 0.7 M), chloride concentration (0 to 80 g/L), source of water (distilled water, seawater and brine), temperature (25 and 45 °C), the sample of concentrate (A and B), nitrate source (analytical grade and industrial salt) and pre-treatments (two different methods).

## 2. Materials and Methods

### 2.1. Concentrate Samples

Two chalcopyrite concentrate samples (namely samples A and B in this study) were obtained from copper mines of the Antofagasta region, Chile. The $P_{80}$ of samples A and B were 105 and 100 μm respectively. The particle size distributions were determined using a Particle Size Analyzer (PSA, Microtrac model S3500 laser diffraction, Verder Scientific, Newtown, PA, USA). The samples were analyzed for chemical composition using atomic absorption spectrometry (AAS, Perkin-Elmer 2380, Perkin Elmer, Wellesley, MA, USA). Copper and iron percentage of the selected samples were 23.8% Cu and 27.5% Fe for sample A, and 25.8% Cu and 23.5% Fe for sample B. As presented in Table 1, the mineralogical composition of the samples was determined using quantitative X-ray diffraction (Siemens/Bruker, Semi-QXRD, model D8 Advance, Germany). Semi-quantitative results were provided by TOPAS (total pattern analysis software) to quantify the sample.

**Table 1.** Mineralogical composition of concentrate samples.

| Chemical Formula | Mineral | Amount (%) | |
| --- | --- | --- | --- |
| | | Sample A | Sample B |
| $CuFeS_2$ | Chalcopyrite | 37.2 | 74.0 |
| CuS | Covellite | 12.5 | - |
| $Cu_9S_5$ | Digenite | 2.9 | - |
| $CuFeO_2$ | Delafossite | 0.8 | - |
| $FeS_2$ | Pyrite | 34.0 | 2.0 |
| $SiO_2$ | Quartz | 12.6 | 2.5 |
| | Gangue | - | 21.5 |
| Total | | 100.0 | 100.0 |

-: this mineral was not found in the sample.

### 2.2. Leaching Solutions

Sulfuric acid ($H_2SO_4$, 95% to 97%, analytical grade), sodium chloride (NaCl, 99.5%, analytical grade), sodium nitrate ($NaNO_3$, 99.5%, analytical grade) and salt of caliche industry, named industrial salt ($NO_3^- = 65.7\%$, $Cl^- = 0.5\%$) were used to prepare leaching solutions. A distilled water, seawater and discard brine (from reverse osmosis) were used as dissolvent. Seawater, from San Jorge Bay, Antofagasta Chile, was pumped and filtered until 1 μm using a polyethylene membrane. Discarded brine was obtained from Coloso's desalination plant, Antofagasta, Chile. The composition of seawater and brine is shown in Table 2.

**Table 2.** Chemical composition of seawater and brine (mg/L).

| Chemical Method | Atomic Absorption Spectrometry | | | | | | Volumetric Analysis | | Gravimetric Analysis |
|---|---|---|---|---|---|---|---|---|---|
| Ionic Species | $Na^+$ | $K^+$ | $Mg^{2+}$ | $Ca^{2+}$ | $Cu^{2+}$ | $NO_3^-$ | $Cl^-$ | $HCO_3^-$ | $SO_4^{2-}$ |
| Seawater | 11,250 | 401 | 1256 | 427 | 0.07 | 2.40 | 20.289 | 149 | 2758 |
| Brine | 19,768 | 746 | 2297 | 355 | 0.14 | 6.40 | 36,074 | 236 | 5063 |

## 2.3. Experimental Procedure

### 2.3.1. Leaching Tests

Jacketed glass reactors of 1 L were used to carry out the stirring leaching tests. Water at desired temperature was circulated by the jacket of the reactor using a thermostatic bath (Julabo bath F25-ME Refrigerated/Heating Circulator). A simplified schematic of the agitated leaching reactor is shown in Figure 1. Mechanical stirring with a Teflon bar was provided to maintain the agitation of the pulp (460 rpm). The solid/liquid ratio used was 50 g of concentrate in 500 mL of solution. The leaching solution was introduced into the reactor. When the desired temperature was achieved, the concentrate sample was put in the reactor and the agitation started. At different time intervals, solution samples (10 mL) were removed from the system for copper analysis using AAS. Redox potential (ORP) and pH values were measured during the test (Hanna portable pH/ORP meter, model HI991003, accuracy ± 0.02 pH y ± 2 mV, Ag/AgCl reference electrode). When the leaching time was achieved (15 days), the agitation stopped, and the pulp was filtered. The solid residue was washed with distilled water and dried at 60 °C until constant weight. Then, the residue was analyzed by AAS for Cu content. The copper extraction rate was calculated using the grades of the head and residual ore, which was corroborated with the results obtained by leaching solutions. A standard deviation of ± 2% was obtained by calculating the copper extraction rates from solid residue and leaching solution for all the tests performed.

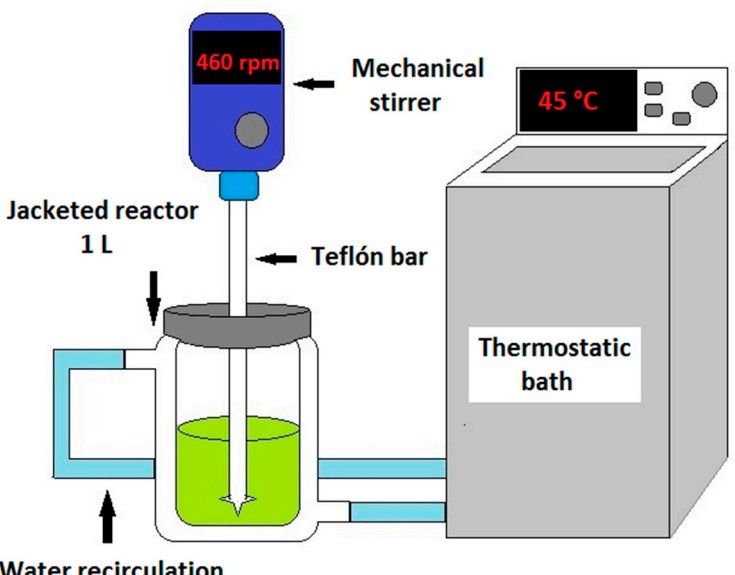

**Figure 1.** Schematic picture of experimental tests.

As shown in the Table 3, different variables were studied: sulfuric acid concentration (0.5 and 0.7 M), sodium nitrate concentration (0.5 and 0.7 M), chloride concentration (0, 20, 36, 40, 60 and 80 g/L), source of water (distilled water, seawater, and brine), temperature (45 and 25 °C), concentrate sample (A and B), nitrate source (analytical grade and industrial salt) and pre-treatment (in two different methods, see Section 2.3.2).

### 2.3.2. Pre-Treatment Tests

Two different pre-treatment methods were carried out in order to improve the kinetic of copper extraction. In the first method of pre-treatment, a solid sample (50 g) was put on the Petri dish. Solid $NaNO_3$ (23 kg/ton of concentrate) and NaCl (12 kg/ton of concentrate) were added and mixed with a spatula. Then, $H_2SO_4$ (17 kg/ton of concentrate) was added to the mix and seawater was sprayed using a sprinkler for even distribution up to a moisture percentage of 15.36%. Petri dish was sealed using a parafilm, and it was left in repose at 25 °C for 20 days. Then, the pretreated solid was leached (test 17, see Table 3). In the second method of pre-treatment, a solution with 0.7 M $NaNO_3$, 0.7 M $H_2SO_4$, and seawater was prepared. A sample of 50 g of concentrate was added into the solution. This pulp was maintained in repose for 20 days without stirring at 25 °C. Then, agitation began for 20 additional days at 25 °C (test 18, see Table 3).

## 3. Results and Discussion

Table 3 shows all leaching tests performed in the different experimental conditions. Maximum copper extraction achieved are presented.

**Table 3.** Leaching tests carried out with concentrate samples.

| N° | $H_2SO_4$ (M) | $NaNO_3$ (M) | T (°C) | Disolvente | Sample | Special Characteristic | Cu (%) |
|----|---------------|--------------|--------|------------|--------|------------------------|--------|
| 1 | 0.7 | 0.7 | 45 | Water | A | - | 70.9 |
| 2 | 0.7 | 0.7 | 45 | Seawater | A | - | 90.6 |
| 3 | 0.7 | 0.7 | 45 | Brine | A | - | 86.6 |
| 4 | 0.7 | 0.7 | 45 | Water | A | $[Cl^-]$ = 20 (g/L) | 93.5 |
| 5 | 0.7 | 0.7 | 45 | Water | A | $[Cl^-]$ = 40 (g/L) | 76.1 |
| 6 | 0.7 | 0.7 | 45 | Water | A | $[Cl^-]$ = 60 (g/L) | 80.4 |
| 7 | 0.7 | 0.7 | 45 | Water | A | $[Cl^-]$ = 80 (g/L) | 88.2 |
| 8 | 0.7 | 0.7 | 45 | Water | A | Industrial salt | 67.2 |
| 9 | 0.7 | 0.7 | 45 | Seawater | A | Industrial salt | 83.5 |
| 10 | 0.7 | 0.7 | 45 | Brine | A | Industrial salt | 77.0 |
| 11 | 0.7 | 0 | 45 | Seawater | A | - | 79.2 |
| 12 | 0.7 | 0.7 | 25 | Seawater | A | - | 60.0 |
| 13 | 0.5 | 0.7 | 45 | Seawater | A | - | 81.9 |
| 14 | 0.7 | 0.5 | 45 | Seawater | A | - | 88.0 |
| 15 | 0.5 | 0.5 | 45 | Seawater | A | - | 76.1 |
| 16 | 0.7 | 0.7 | 45 | Seawater | B | - | 80.8 |
| 17 | 0.7 | 0.7 | 25 | Seawater | A | Pre-treatment (Method 1) | 63.8 |
| 18 | 0.7 | 0.7 | 25 | Seawater | A | Pre-treatment (Method 2) | 71.4 |

-: this means that no special characteristic present this test.

### 3.1. Effect of Nitrate Concentration

Figure 2a shows that the increase of nitrate concentrations from 0, 0.5, and 0.7 M increases the copper extraction from 79.2%, 88.0%, and 90.6%, respectively, after 15 days. Increasing nitrate concentration in the leaching medium improves copper extraction. This has been demonstrated by previous studies [20,25,26]. Nitrate can be provided by caliche's industry as discard salts or discard/intermediate solutions. The nitrate-acid medium help the dissolution of copper from sulfides [20,29]. This system can form elemental sulfur or natrojarosite as solid residue [31,32], high temperatures or pressures would not be necessary, and the oxidation-reduction potential of the reaction is high. The pH values were maintained at ≤1.21 for all tests performed. Figure 2b shows the redox potential values of the leaching tests. ORP were in ranges between 890.8 to 920.8, 933.8 to 965.8, and 942.3 to 983.3 mV vs. SHE for 0, 0.5, and 0.7 M $NaNO_3$, respectively. These high potentials demonstrate that the system is highly oxidizing.

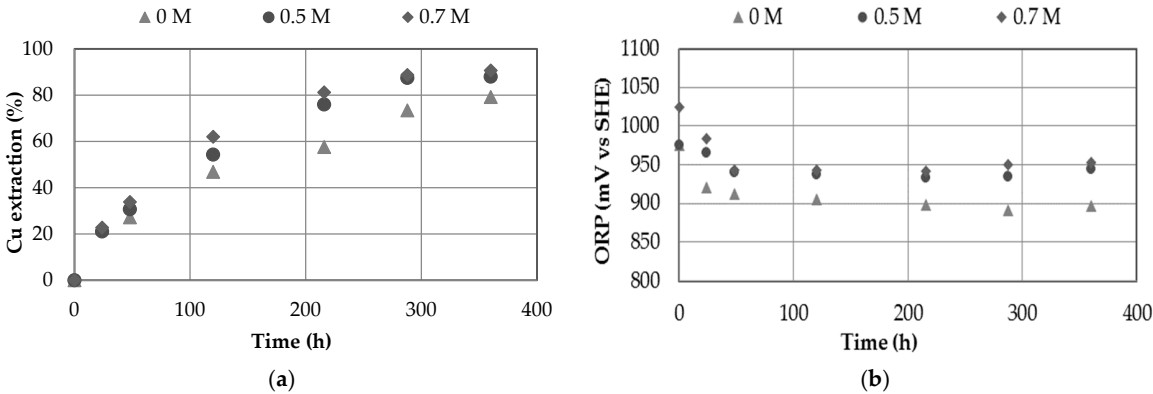

**Figure 2.** (**a**) Copper extraction (%) vs. time (h) at different sodium nitrate concentration: 0, 0.5 and 0.7 M. (**b**) Redox potential (mV) vs. time (h) at different sodium nitrate concentration: 0, 0.5 and 0.7 M. Experimental conditions: $[H_2SO_4]$ = 0.7 M, seawater as a dissolvent, 45 °C and sample A.

### 3.2. Effect of Sulfuric Acid Concentration

Figure 3a shows that the increase of sulfuric acid concentrations from 0.5 to 0.7 M increases the copper extraction from 81.9% to 90.6%, after 15 days. The effect of increasing sulfuric acid concentration showed better copper extraction in comparison with the increase of sodium nitrate concentration (Figure 2a), where copper extraction slightly improved. Castellón et al. [21] showed that the variable sulfuric acid concentration is more preponderant in the system than the nitrate concentration using the analysis of variance (ANOVA). The pH values were maintained at ≤1.46 for all tests conducted. Figure 3b shows the redox potential values for the tests performed. ORP were in the ranges between 933.8 to 954.1 and 942.3 to 983.3 mV vs. SHE for 0.5 and 0.7 M $H_2SO_4$, respectively.

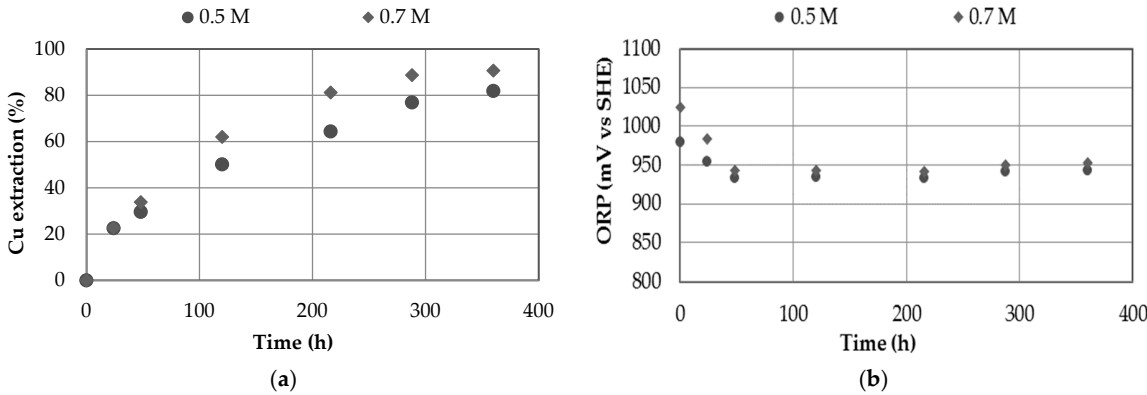

**Figure 3.** (**a**) Copper extraction (%) vs. time (h) at different sulfuric acid concentration: 0.5 and 0.7 M. (**b**) Redox potential (mV) vs. time (h) at different sulfuric acid concentration: 0.5 and 0.7 M. Experimental conditions: $[NaNO_3]$ = 0.7 M, seawater as dissolvent, 45 °C and sample A.

### 3.3. Different Equivalent Sodium Nitrate and Sulfuric Acid Concentration

Figure 4a shows that the increase of equivalent sodium nitrate and sulfuric acid concentration from 0.5 to 0.7 M increases the copper extraction from 76.1% and 90.6 %, respectively, after 15 days. The pH values were maintained at ≤1.58 for all tests performed. Figure 4b shows the redox potential values of the leaching tests conducted. ORP were in the ranges between 907.8 to 940.8 and 942.8 to 983.3 mV vs. SHE for 0.5 and 0.7 M of both reagents, respectively. Castellón et al. [21] studied the concentrate leaching in the nitrate-chloride-acid media. The test at 0.5 M $NaNO_3$ + 0.5 M $H_2SO_4$ in seawater was carried out where the maximum of 90.8% copper extraction was achieved after 94 h using a concentrate with $P_{80}$ of 60.7 μm. The differences could be attributed to the particle size of the samples and the mineralogy of the concentrate. In the study of Castellón et al. [21], the concentrate

sample had a higher chalcopyrite presence (61.5%), covellite (1.5%) and chalcanthite (1.2%) compared to the current work (see Table 1).

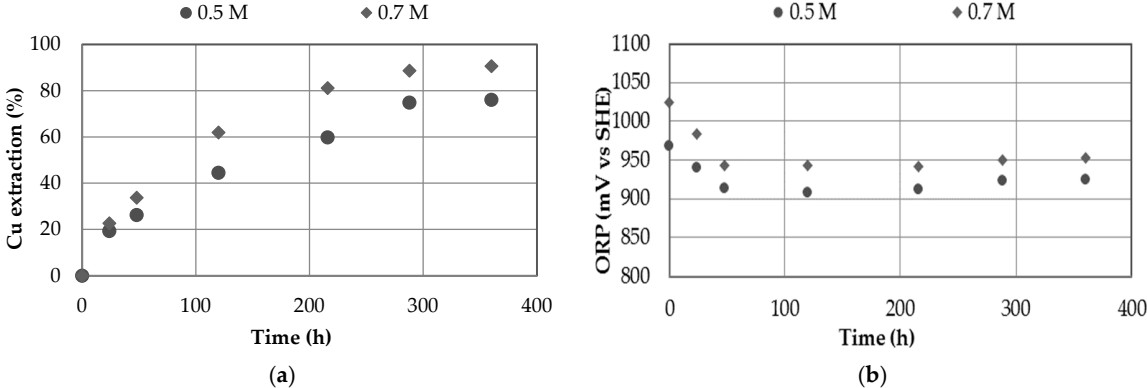

**Figure 4.** (**a**) Copper extraction (%) vs. time (h) at different equivalent sodium nitrate and sulfuric acid concentration: $[NaNO_3] = [H_2SO_4] = 0.5$ and 0.7 M. (**b**) Redox potential (mV) vs. time (h) at different sulfuric acid concentration: 0.5 and 0.7 M. Experimental conditions: seawater as dissolvent, 45 °C and sample A.

### 3.4. Effect of Dissolvent

Figure 5a shows that the seawater system obtained better copper extraction (90.6%) than brine (86.6%) or distilled water (70.9%) based leaching media after 15 days. Clearly, when chloride ions are present in the media, provided by seawater or brine; the copper extraction increase in comparison when these ions are not present in the distilled water system. The pH values were maintained at ≤1.66 for all tests conducted. Figure 5b shows the redox potential values for the leaching tests performed. ORP were in the ranges between 885.8 to 915.8, 942.3 to 983.3, and 945.8 to 967.8 mV vs. SHE for distilled water, seawater, and brine systems, respectively. In a chloride media, the redox potential is higher than when distilled water is used. As reported in the literature, the use of sodium chloride forms complexes with Cu ions, avoiding passivation of the mineral surface, stabilizing cuprous ions and in turn increasing the solubility of the metal. It would be expected that the dissolvent from a reverse osmosis process (brine) would obtain better recovery rates due to high chloride concentration, but an excess of other ions in the solution could be affecting the copper dissolution. It should be explained by the high ionic strength present in the leaching media due to the presence of ions from brine, nitrate and the acid added to the system. This effect has been observed in other systems (Figure 2 of [33], Figure 8 of [32]).

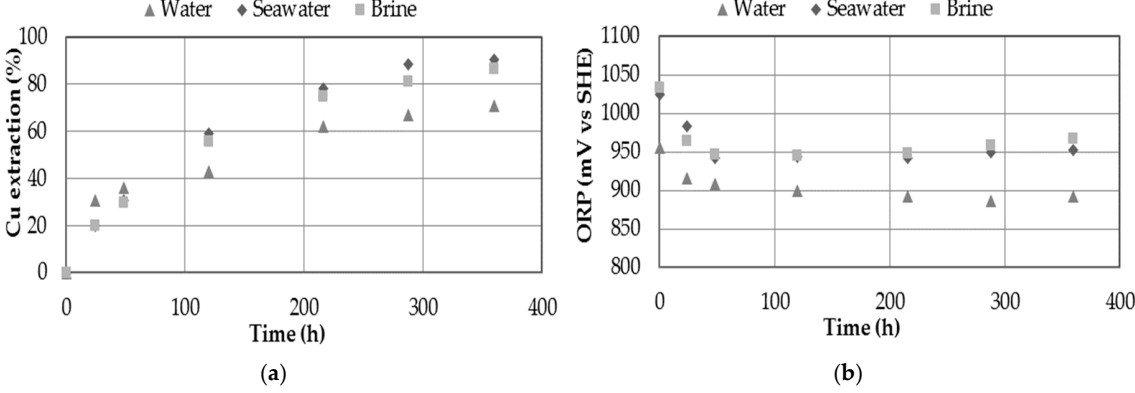

**Figure 5.** (**a**) Copper extraction (%) vs, time (h) at different water sources: water, seawater and brine. (**b**) Redox potential (mV) vs. time (h) at different water sources: water, seawater and brine. Experimental conditions: $[NaNO_3] = [H_2SO_4] = 0.7$ M, 45 °C and sample A.

## 3.5. Effect of Chloride Concentration

Figure 6a shows that for high chloride concentrations of 60 and 80 g/L, the copper leaching kinetics are fast in the first 120 h, which is in agreement with the literature [34–36], but over time, the leaching kinetics decrease. The 20 g/L Cl⁻ system obtained the maximum extraction of 93.5% Cu, followed by 80 g/L Cl⁻ with 88.2% Cu; 60 g/L Cl⁻ with 80.4% Cu; 40 g/L Cl⁻ with 76.1% Cu and 0 g/L Cl⁻ with 70.9% Cu. An increase in chloride concentration is not necessary to increase the copper extraction in the studied system. This was observed by other studies [32,33]. This agrees with the result shown in Figure 5a, where the seawater system is much more efficient in copper extraction than the system using brine. The pH values were maintained at ≤1.72 for all tests conducted. Figure 6b shows the redox potential values of the tests. ORP were in the ranges between 885.8 to 915.8, 933.8 to 969.6, 949.8 to 969.3, 957.8 to 976.3, and 969.8 to 988.3 mV vs. SHE for 0, 20, 40, 60, and 80 g/L Cl⁻, respectively. An increase in the redox potential was observed when increasing the chloride concentration. This behavior was observed by Torres et al. [9].

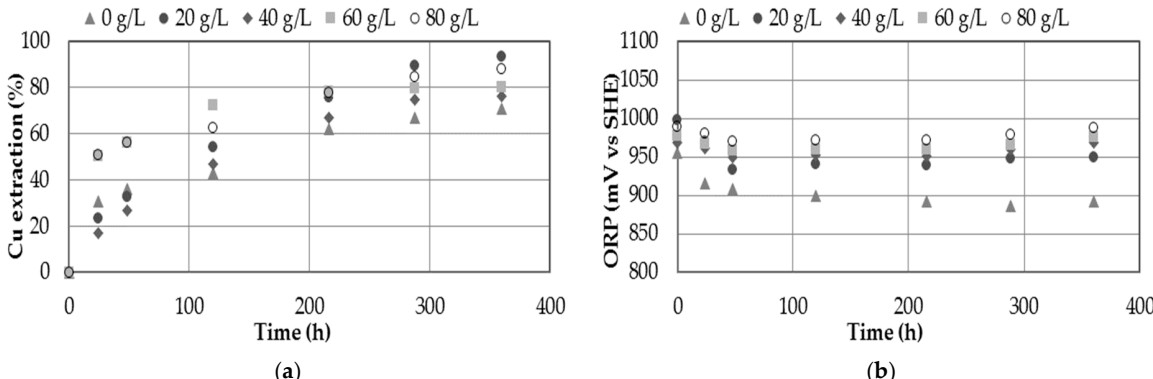

**Figure 6.** (**a**) Copper extraction (%) vs. time (h) at different chloride concentration: 0, 20, 40, 60 and 80 g/L. (**b**) Redox potential (mV) vs. time (h) at different chloride concentration: 0, 20, 40, 60 and 80 g/L. Experimental conditions: [NaNO₃] = [H₂SO₄] = 0.7 M, water as dissolvent, 45 °C and sample A.

Figure 7a shows the comparison between the system using seawater as dissolvent and the system using distilled water and 20 g/L Cl⁻. According to the figure, both curves are similar. This confirms that the chloride ion is the ion that would be influencing the leaching system with seawater and not the other ions present. This agrees with what Hernandez et al. observed (Figure 8 in [20]). Figure 7b shows a similar behavior of redox potential values in both systems.

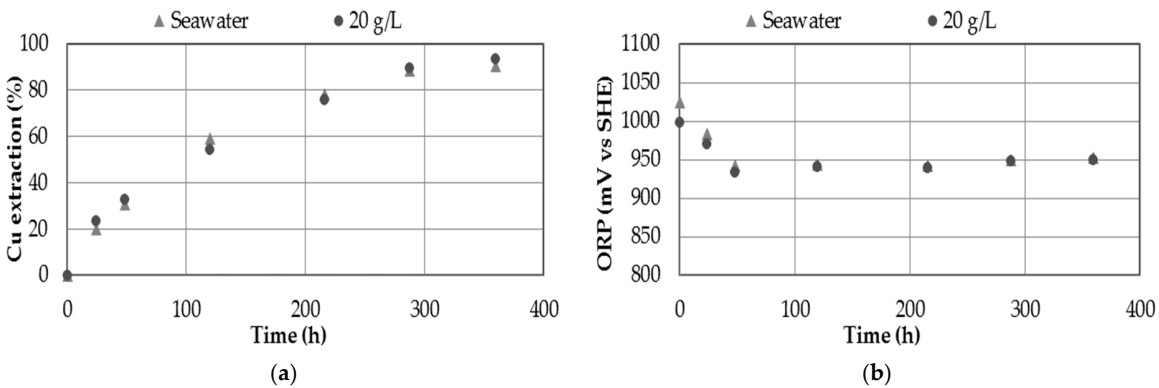

**Figure 7.** (**a**) Comparison of copper extraction (%) curve when seawater and water with 20 g/L Cl⁻ were used as a source of water in the leach solution. (**b**) Redox potential (mV) versus time (h) when seawater and water with 20 g/L Cl⁻ were used as a source of water in the leach solution. Experimental conditions: [NaNO₃] = [H₂SO₄] = 0.7 M, 45 °C and sample A.

Figure 8 shows the comparison between the system using seawater and brine as dissolvent and the systems using distilled water with 20 and 40 g/L Cl⁻. According to the figure, the system at 40 g/L Cl⁻ shows a lower copper extraction, increasing when brine was used, followed by seawater or water with 20 g/L Cl⁻.

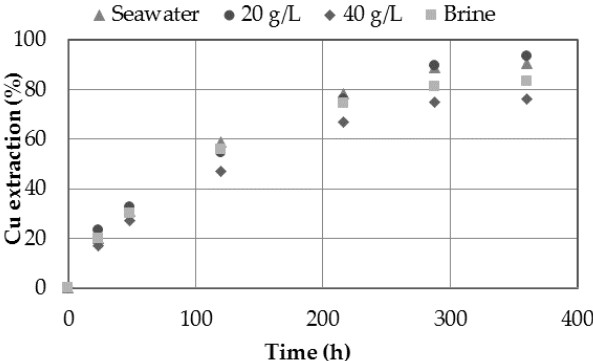

**Figure 8.** Comparison of copper extraction (%) curve when seawater, brine, water with 20 g/L Cl⁻ and water with 40 g/L Cl⁻ were used as source of water in leach solution. Experimental conditions: [NaNO₃] = [H₂SO₄] = 0.7 M, 45 °C and sample A.

### 3.6. Effect of Temperature

As observed in Figure 9a, the temperature has a strong influence on copper leaching kinetics, as reported in the literature [25,26,37]. Copper extraction increases with increasing temperature. At 25 °C, 42.4% Cu was extracted after 360 h of leaching. Compared to 45 °C, where 90.6% Cu was extracted in the same leaching time. As shown in the plot, by increasing the leaching time from 360 to 960 h, at 25 °C, the copper extraction only increased from 42.4% until 60.0%. In both systems, the kinetic curves tend to decrease. This decrease in slopes is due to the formation of a passivating layer, which could be sulfur or natrojarosite, according to what is reported in the literature [4,16,26,31]. Figure 9b shows the redox potential values, which were in the ranges between 885.8 to 924.8, and 942.3 to 983.3 mV vs. SHE at 25 and 45 °C, respectively.

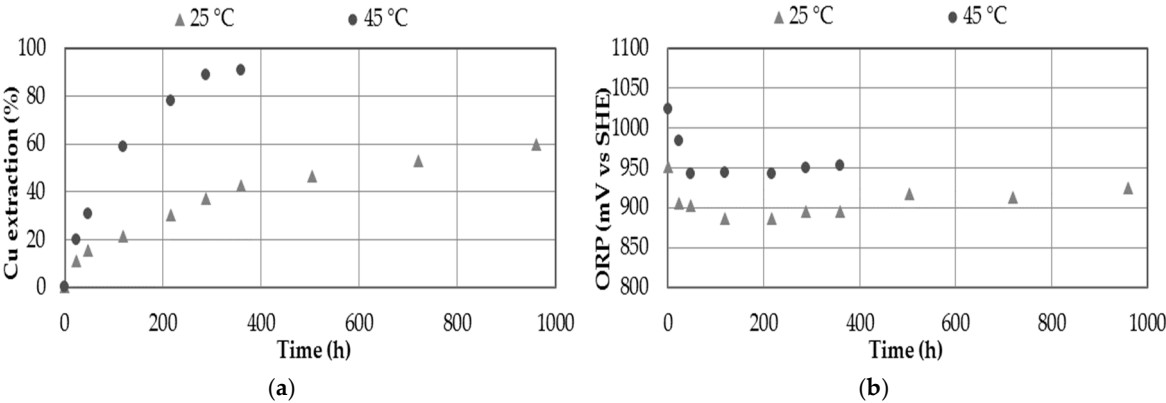

**Figure 9.** (**a**) Copper extraction (%) vs. time (h) at different temperature: 25 and 45 °C. (**b**) Redox potential (mV) vs. time (h) at different temperature: 25 and 45 °C. Experimental conditions: [NaNO₃] = [H₂SO₄] = 0.7 M, seawater as dissolvent and sample A.

### 3.7. Effect of Using Industrial Salt as a Source of Nitrate

Figure 10a shows the difference when having the same amount of 0.7 M nitrate concentration using industrial and analytical salt, reaching a maximum of 90.6% for analytical grade and 67.2% for industrial salt. This decrease in extraction could be mainly due to the impurities present in the

industrial salt that would affect the extraction of copper. Likewise, it provides an alternative to be used, where the caliche industry can supply raw materials to the copper industry. Figure 10b shows the redox potential values which were in the ranges between 942.3 to 983.3 and 922.8 to 980.8 mV vs. SHE when analytical and industrial salt were used.

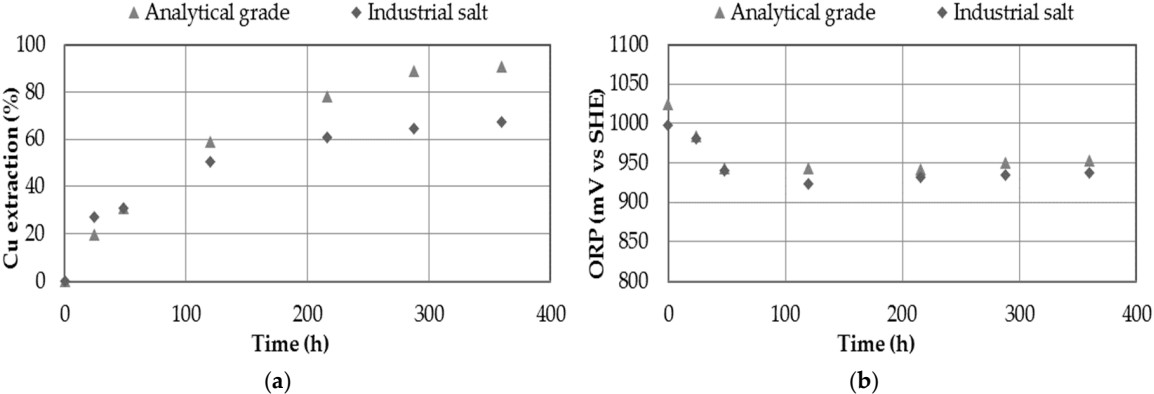

**Figure 10.** (**a**) Copper extraction (%) vs. time (h) when different nitrate sources were used: analytical grade and industrial salt. (**b**) Redox potential (mV) vs. time (h) when different nitrate sources were used: analytical grade and industrial salt. Experimental conditions: $[NaNO_3]$ = $[H_2SO_4]$ = 0.7 M, 45 °C, seawater as dissolvent and sample A.

Figure 11a shows the curves of copper extraction with increasing chloride concentration using three types of dissolvent, distilled water, seawater and brine including the addition of industrial salt as a source of nitrate ions. The copper extraction percentages obtained were 67.2%, 83.5%, and 77.0%, respectively. It can be observed that the system with seawater and industrial salt shows better copper extraction than the other two systems. This behavior is similar to the trend observed when the dissolvent effect was analyzed (without the addition of industrial salt as a source of nitrate ions), where the system with seawater showed better efficiencies in the copper solution than the system with brine. Figure 11b shows the redox potential values, which were in the ranges between 922.8 to 980.8, 945.8 to 965.8 and 943.8 to 983.8 mV vs. SHE when water, seawater and brine were used with industrial salt.

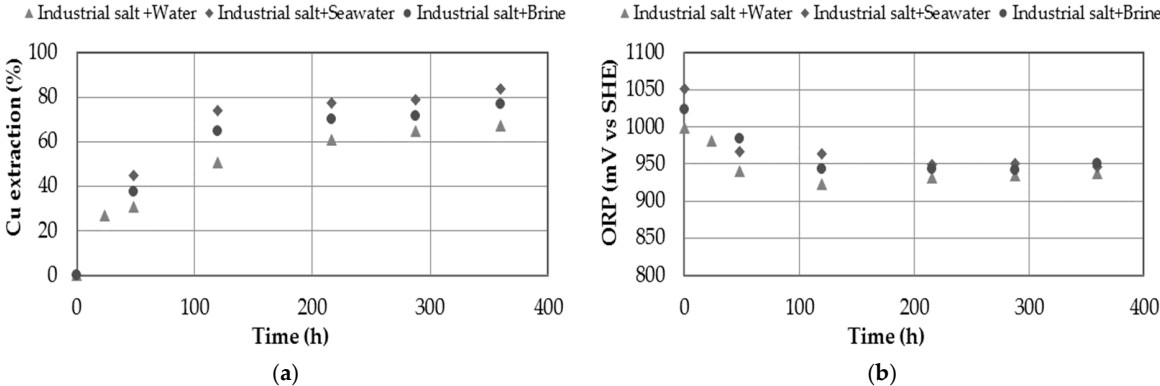

**Figure 11.** (**a**) Copper extraction (%) vs. time (h) when different sources of water were used with industrial salt. (**b**) Redox potential (mV) vs. time (h) when different sources of water were used with industrial salt. Experimental conditions: $[NaNO_3]$ = $[H_2SO_4]$ = 0.7 M, 45 °C and sample A.

### 3.8. Effect of Concentrate Grade

As can be seen in Figure 12a, concentrate A obtained higher copper extraction values during the test, reaching a maximum of 90.6% Cu, compared to concentrate B with a maximum of 80.8% Cu. According to the mineralogy of both samples, it can be seen that concentrate B has a high

percentage of chalcopyrite compared to concentrate A. Furthermore, concentrate A shows the presence of other copper sulfides that could be leached faster than chalcopyrite, providing more copper ions to the system. Concentrate A has a high percentage of pyrite that could be leached according to the following reactions:

$$FeS_2 + NO_3^- + 4H^+ = Fe^{3+} + NO + 2H_2O + 2S \qquad \Delta G_{45\,°C} = -31.8\ kcal \qquad (1)$$
$$3FeS_2 + 2NO_3^- + 8H^+ + 6Cl^- = 3FeCl_2 + 2NO + 4H_2O + 6S \qquad \Delta G_{45\,°C} = -24.1\ kcal \qquad (2)$$

Reaction (1) generates the ferric ion that oxidizes the mineral, allowing for a greater dissolution of copper and iron. Another relevant factor is the high presence of gangue in sample B (Table 2), which hinders the extraction of copper and iron. Figure 12b shows the redox potential values, which were in the ranges between 942.3 to 983.3, and 927.3 to 968.3 mV vs. SHE when A or B samples were used, respectively.

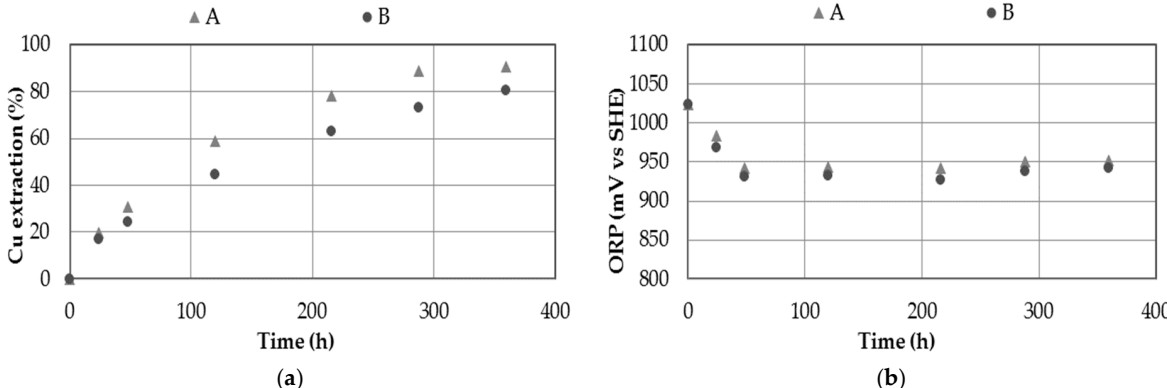

**Figure 12.** (**a**) Copper extraction (%) vs. time (h) when different concentrate samples were used: A and B. (**b**) Redox potential (mV) vs. time (h) when different concentrate samples were used: A and B. Experimental conditions: $[NaNO_3] = [H_2SO_4] = 0.7$ M, 45 °C and seawater as dissolvent.

### 3.9. Effect of Pre-Treatment

The results show that pre-treatment has a positive effect on copper extraction after 40 days, as shown in Figure 13a. The copper extraction percentages obtained are 63.8%, 71.4% and 60.0% Cu in the systems with pre-treatment method 1, pre-treatment method 2 and without pre-treatment, respectively. The pre-treatment, known as curing, accelerates the dissolution of concentrate due to contact with solid particles and oxidant-acid solution (sulfuric acid concentrate and sodium nitrate-seawater) with low moisture of the sample (≤15%). During the repose time, dissolution reaction occurs on the surface of the concentrate. This was studied by several researchers with positive effects on copper dissolution [7,8,29]. Better copper extraction was obtained using pre-treatment method 2 in comparison with pre-treatment method 1. This can be attributed to the fact that in pre-treatment method 2, the solution was in contact with the concentrate for 20 days, with high sulfuric acid concentration, nitrate, and chloride ions. Whereas in the approach of pre-treatment method 1, the process was semi-dry because the moisture of the sample was only 15 %, and the sample lost moisture in the time.

In any case, it is necessary to carry out an economic study of these processes to compare the cost of leaching by agitation of a concentrate at room temperature for 40 days versus pretreating the concentrate and leaving it in semi-dry rest or in a leaching solution for 20 days and then leach for an additional 20 days. Both pre-treatment methods provide a possible alternative to extract copper from concentrates. Figure 13b shows the redox potential values that were in the ranges between 820.8 to 875.8, 842.7 to 857.7, and 878.7 to 895.7 mV vs. SHE in tests without pre-treatment, pre-treatment method 1, and pre-treatment method 2, respectively.

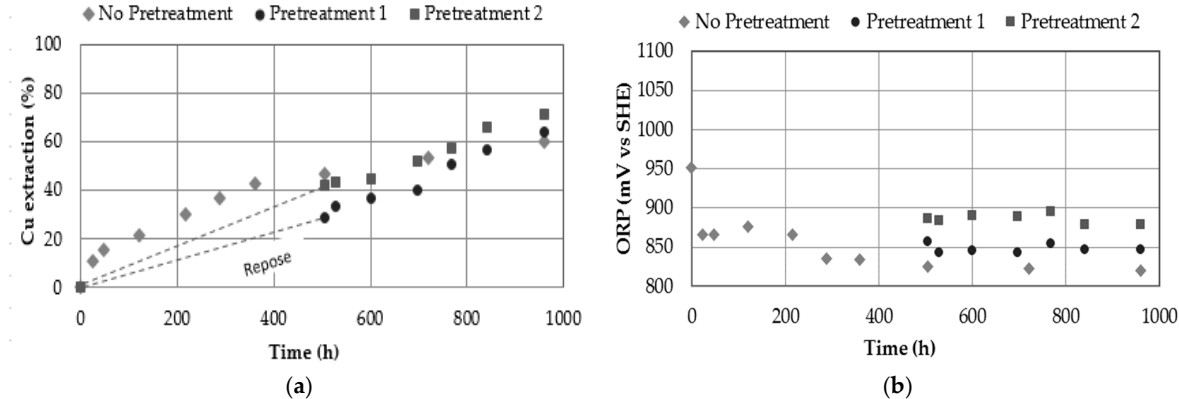

**Figure 13.** (**a**) Copper extraction (%) vs. time (h) at different pre-treatment. (**b**) Redox potential (mV) vs. time (h) at different pre-treatment. Experimental conditions: [NaNO$_3$] = [H$_2$SO$_4$] = 0.7 M, 25 °C, seawater as dissolvent and sample A.

## 4. Conclusions

This study aims to introduce an alternative method to confront the problems that distress Chilean copper mining such as the depletion of oxidized copper minerals and the shortage of water. A green and environmentally friendly hydrometallurgical approach for chalcopyrite concentrate leaching was studied. In the core of the circular economy and industrial waste valorization, seawater/brine (as a source of chloride) and caliche's salts (as a source of nitrate ions) were used. Chalcopyrite leaching was studied in a nitrate-chloride-acid medium where the use of seawater or brine provides the chloride ion in leaching. The main conclusions are the following:

1. Increasing the nitrate concentration from 0 to 0.7 M showed positive effects on copper extraction, although the most significant effect was observed by increasing the concentration of sulfuric acid from 0.5 to 0.7 M.

2. The presence of chloride in the medium showed increasing on copper extraction. The presence of chloride ions can be provided through the use of seawater, discard brines from desalination plants or by adding sodium chloride, obtaining a maximum extraction of 93.5% Cu when 20 g/L Cl$^-$ was used.

3. The effect of temperature significantly influences dissolution kinetics of copper, reaching 90.6% Cu at 45 °C after 15 days of leaching in comparison to 42.4% Cu at 25 °C in the same conditions.

4. The use of industrial salt providing 0.7 M of nitrate in a leaching medium with 0.7 M of sulfuric acid did not obtain better extraction percentages than using analytical salt, due to the presence of other impurities that affect the dissolution system. In any case, it is a possible alternative to evaluate, obtaining percentages of copper extraction of 67%.

5. The leaching of two samples of concentrates showed different copper extraction values, mainly due to the mineralogical and chemical composition of each concentrate. Sample A has a greater presence of pyrite, which could contribute ferric ion to the medium, which would help leach chalcopyrite, as this is an oxidizing ion.

6. The use of pre-treatment for 20 days produces extractions close to 28.4% and 42.3% Cu, using pre-treatment 1 and pre-treatment 2 at 25 °C, respectively. Copper extraction can increase to values close to 63.8 and 71.4% by leaching these pretreated concentrates for an additional 20 days. A leach at 25 °C for 40 days obtained a 60% copper extraction. It is necessary to evaluate the cost of pretreating the concentrate and leaving it at rest, versus leaching it.

These results show that the nitrate-chloride-acid system presents good copper dissolution results at moderate temperatures of ≤45 °C and atmospheric pressure. This alternative must be evaluated at an industrial level using other sources of raw materials, such as process brines (e.g., from desalination

plants), raw seawater, seawater from thermoelectric plants, process waters that currently consider as waste, as well as salts and solutions of discards or intermediate processes of the caliche industry.

**Author Contributions:** Conceptualization, P.H. and A.D.; methodology, A.D.; investigation, A.D., validation, N.T., Y.G. and J.C.; formal analysis, P.H., M.M., N.T., J.C. and Y.G.; investigation, P.H. and A.D., writing—original draft, P.H. and M.M., writing-review & editing, Y.G. All authors have read and agreed to the published version of the manuscript.

**Funding:** This research was funded by ANID-Chile through Fondecyt de Iniciación Project N°11170179 and Project ING2030 CORFO Code 16ENI2–71940.

**Acknowledgments:** The authors thank to Laboratorio de Investigación de Procesos of Universidad de Antofagasta for the support provide.

**Conflicts of Interest:** The authors declare no conflict of interest.

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
