# Peer review of "Use of Seawater/Brine and Caliche’s Salts as Clean and Environmentally Friendly Sources of Chloride and Nitrate Ions for Chalcopyrite Concentrate Leaching"

_minerals, doi:10.3390/min10050477_

Round 1

Reviewer 1 Report

The paper is well presented and the experimental results would be useful for industrial point of view. 

My main recommendations are:

minor word changes:

19 please change the words "environmentally friendlier" for  less harmful for the environment

30 cost effective is a better way to describe this. 

32  pre-treatment

I think the introduction is not clear or well organized, the aim of the project is clear but the gaps in knowledge and specific problems are not well defined. Please reorganize the introduction in a more logical manner. 

  • Please review the introduction order it should present the problem, the available options and research and the proposed research objective.
  • The problems associated to chalcopyrite leaching under regular conditions for pyrometallurgical and hydrometallugircal methods are not well defined. 
  • Please make clear why are you trying the fluids selected maybe including some information about the Chilean government  proposal for sustainable mining and the some numbers about the presence of chalcopyrite and the low availability of water.
  • Several works have been presented with nitrate and seawater leaching of chalcopyrite please explain what is new in your research. 

You concluded that mineralogical changes are the reason for different dissolution at different concentrates. However, I would like to see the XRD and the QEMSCAN information to compare and support this conclusion. You have the initial mineralogy but no XRD pattern. You also suggest the formation of natrojarosite, it would be useful to have an image of after leaching residue in SEM to confirm if any product layer is formed.

Please be careful when you talk about "green" alternative, you are still using sulfuric acid. (e.g., even pure water addition to an orebody could disturb the existing environment, so no lixiviant is truly ‘green’), better to say more environmentally friendly. 

Author Response

We uploaded a file with the answers.

Reviewer 2 Report

The paper can be printed in the journal after a minor revision. Remarks are given in the paper.

Author Response

We uploaded a file with the response to reviewer 2.

Thanks.
